# Multi-Session Budget Optimization for Forward Auction-based Federated Learning

**Xiaoli Tang**[1]   **Han Yu**[1]   **Zengxiang Li**[1]   **Xiaoxiao Li**[2,3]

## Abstract

Auction-based Federated Learning (AFL) has emerged as an important research field in recent years. The prevailing strategies for FL data consumers (DCs) assume that the entire team of the required data owners (DOs) for an FL task must be assembled before training can commence. In practice, a DC can trigger the FL training process multiple times. DOs can thus be gradually recruited over multiple FL model training sessions. Existing bidding strategies for AFL DCs are not designed to handle such scenarios. Therefore, the problem of multi-session AFL remains open. To address this problem, we propose the Multi-session Budget Optimization Strategy for forward Auction-based Federated Learning (`MBOS-AFL`). Based on hierarchical reinforcement learning, `MBOS-AFL` jointly optimizes inter-session budget pacing and intra-session bidding for AFL DCs, with the objective of maximizing the total utility. Extensive experiments on six benchmark datasets show that it significantly outperforms seven state-of-the-art approaches. On average, `MBOS-AFL` achieves 12.28% higher utility, 14.52% more data acquired through auctions for a given budget, and 1.23% higher test accuracy achieved by the resulting FL model compared to the best baseline. To the best of our knowledge, it is the first budget optimization decision support method with budget pacing capability designed for DCs in multi-session forward AFL.

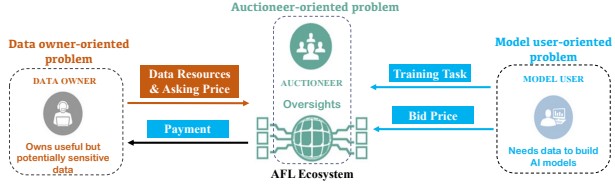

*Figure 1.* An overview of auction-based federated learning (AFL).

## 1. Introduction

Federated Learning (FL) (Fan et al., 2025; Rendle, 2012; Yang, 2020; Meng et al., 2024) has gained great attention for its ability to safeguard data privacy and user confidentiality in both academia (Qi et al., 2025b;a; Tang et al., 2024c) and industry (Sun et al., 2024). Traditional FL approaches often assume that data owners (DOs), also known as FL clients, are willing participants in FL tasks, assisting data consumers (DCs), or FL servers, in training models. However, this assumption may not always hold in practice, as DOs often weigh their participation against self-interest and cost-benefit considerations. To address this challenge, auction-based Federated Learning (AFL) has emerged as a promising solution (Jiao et al., 2019; Deng et al., 2021; Zhang et al., 2021; He et al., 2024).

As shown in Fig. 1, the key participants in AFL are the auctioneer, DOs, and DCs. The data trading process between DOs and DCs is modeled as an auction, coordinated by the auctioneer. The auctioneer facilitates the flow of asking prices from DOs and bid prices from DCs. After receiving the bids, the auctioneer consolidates the results, notifies participants of the match-making outcomes, and determines the auction winners. Through these auction processes, DCs recruit DOs for FL training tasks. Once FL teams (i.e., DOs recruited for the FL training tasks) are formed, DCs proceed with the FL model training following standard FL protocols.

AFL methods can be divided into three categories (Tang et al., 2024a; Tang & Yu, 2023c): 1) data owner-oriented (DO-oriented), 2) auctioneer-oriented, and 3) data consumer-oriented (DC-oriented). DO-oriented AFL methods focus on helping DOs determine the amount of resources to com-

---

[1]College of Computing and Data Science, Nanyang Technological University, Singapore [2]Department of Electrical and Computer Engineering, The University of British Columbia, Canada [3]Vector Institute, Canada. Correspondence to: Han Yu <han.yu@ntu.edu.sg>.

*Proceedings of the 42nd International Conference on Machine Learning*, Vancouver, Canada. PMLR 267, 2025. Copyright 2025 by the author(s).

mit to FL tasks, and set their respective reserve prices for profit maximization. Auctioneer-oriented AFL methods investigate how to optimally match DOs with DCs as well as provide the necessary governance oversight to ensure desirable operational objectives can be achieved (e.g., fairness, social cost minimization (Zhang et al., 2023)). DC-oriented AFL methods examine how to help DCs select which DOs to bid and for how much, in order to optimize key performance indicators (KPIs) within budget constraints, possibly in competition with other DCs.

This paper focuses on DC-oriented AFL, helping DCs bid for DOs. The prevailing methods in this domain require that the budget of a DC shall be maximally spent to recruit the entire team of necessary DOs before FL model training can commence (Tang & Yu, 2023b; Tang et al., 2024b; Tang & Yu, 2023a). In practice, throughout the FL model training process, a DC can recruit DOs over multiple training sessions. This is especially useful in continual FL (Yoon et al., 2021; Pan et al., 2016) settings where DOs' local data are continuously updated over time. Existing AFL approaches designed to optimize KPIs within a single auctioning session cannot be directly applied in multi-session AFL scenarios, especially in scenarios with multiple DCs competing to bid for DOs from a common pool of candidates. This is primarily due to the limitation that they are unable to perform budget pacing, which pertains to the strategic dispersion of a limited overall budget across multiple AFL sessions to achieve optimal KPIs over a given time frame.

To address this issue, we propose the Multi-session Budget Optimization Strategy for forward Auction-based Federated Learning (`MBOS-AFL`). It is designed to empower a DC with the ability to dynamically allocate its limited budget over multiple AFL DO recruitment sessions, and then optimize the distribution of budget for each session among DOs through effective bidding. The ultimate goal is to maximize the DC's winning utility. `MBOS-AFL` is grounded in Hierarchical Reinforcement Learning (HRL) (Pateria et al., 2021) to effectively deal with the intricate decision landscape and the absence of readily available analytical remedies. Specifically, `MBOS-AFL` consists of two agents for each DC: 1) the Inter-Session Budget Pacing Agent (`InterBPA`), and 2) the Intra-Session Bidding Agent (`IntraBMA`). For each auctioning session, each DC's `InterBPA` opportunistically determines how much of the total budget shall be spent in this session based on jointly considering the quantity and quality of the currently available candidate DOs, as well as bidding outcomes from previous sessions. Then, the DC's `IntraBMA` determines the bid price for each data resource offered by DOs in the AFL market within the session budget.

To our best knowledge, `MBOS-AFL` is the first budget optimization decision support method with budget pacing capability designed for DCs in multi-session forward auction-

based FL. Extensive experiments on six benchmark datasets show that it significantly outperforms seven state-of-the-art approaches. On average, `MBOS-AFL` achieves 12.28% higher utility, 14.52% more data acquired through auctions for a given budget, and 1.23% higher test accuracy achieved by the resulting FL model compared to the best baseline.

## 2. Related Work

Existing methods for DC-oriented issues can be further divided into two subcategories (Tang, 2024; Tang & Yu, 2025): i) reverse auction-based methods, and ii) forward auction-based methods.

**Reverse Auction-based Methods**: Developed primarily for monopoly AFL markets where there is only one DC facing multiple DOs, reverse auction-based methods like (Deng et al., 2021; Zhang et al., 2021; Jiao et al., 2020; Zeng et al., 2020; Tang & Yu, 2024c; Le et al., 2020; Thi Le et al., 2021) address the challenge of DO selection through reverse auctions. The key idea of these methods is to optimally resolve the DO selection problem, targeting the maximization of KPIs specific to the target DC. Particularly relevant in scenarios where disparate DOs vie for the attention of a sole DC, these methods have progressed by integrating diverse mechanisms such as graph neural networks, blockchains, and reputation assessment.

**Forward Auction-based Methods**: These methods are designed for situations where multiple DCs compete for the same pool of DOs (Tang & Yu, 2023b; 2024a; 2023a; 2024b; Tang et al., 2024b). The key idea of these methods lies in determining the optimal bidding strategy for DCs. The goal is to maximize model-specific key performance indicators. A notable example is Fed-Bidder (Tang & Yu, 2023b) which assists DCs to determine their bids for DOs. It leverages a wealth of auction-related insights, encompassing aspects like DOs' data distributions and suitability to the task, DCs' success probabilities in ongoing auctions and budget constraints. However, this method ignores the complex relationships among DCs, which are both competitive and cooperative. To deal with this issue, (Tang & Yu, 2023a) models the AFL ecosystem as a multi-agent system to steer DCs to bid strategically toward an equilibrium with desirable overall system characteristics.

`MBOS-AFL` falls into the forward auction-based methods category. Distinct from existing methods which focus on optimizing the objectives within a single auctioning session, it is designed to solve the problem of multi-session AFL budget optimization.

# 3. Preliminaries

**AFL Market**: Generally, an AFL market consists of three types of participants (Tang et al., 2024a): 1) Data Owners (DOs): entities possessing potentially sensitive yet valuable data, who are willing to share or sell access to their data resources for FL task training in exchange for appropriate compensation. 2) Data Consumers (DCs): organizations or individuals requiring data to train their machine learning models via FL. 3)Auctioneer: a trusted third-party entity orchestrating the auction process between DOs and DCs. It facilitates the exchange of data resources for FL training tasks through an auction mechanism, such as the Second-Price Sealed-Bid (SPSB) auction (Yang et al., 2023).

When a DO is ready to offer its services for FL task training, it notifies the auctioneer, specifying its bid request and the reserve price.[1] The auctioneer then announces the auction to all DCs currently participating in the AFL market. Any DC whose required the corresponding data resources aligns with the DO's offering submits a bid for the auction.

**Multi-Session Budget Constrained AFL Bidding**: During the course of FL model training, a DC can initiate the FL training procedure (i.e., a *training session*) on multiple occasions, with the aim of recruiting DOs to improve model performance. Consider the scenario of multiple banks engaging in FL. The dynamic nature of user data within these banks sets in motion a perpetual cycle of updates, with continually refreshed data stored locally by each bank. As a result, these banks systematically engage in repeated sessions of federated model training periodically, during which the standard FL training protocol is followed. Let $S$ denote the number of training sessions for the target DC, who has a budget $B$ for all training sessions $[S]$. In each FL training session $s$ ($s \in [S]$), there are $C_s$ available qualified DOs, which can help train the FL model of the target DC. Each DO $i \in [C_s]$ possesses a private dataset $D_i = \{(\boldsymbol{x}_j, y_j)\}_{j=1}^{|D_i|}$.

Following (Tang & Yu, 2023b), we assume that each DO $i$ become gradually available over time. Each DO $i$ can trigger the following auction process: 1) **Bid Request Initiation**: DO $i \in [C_s]$ generates a bid request about itself (e.g., identity, data quantity, etc.) and sends it along with the the reserve price (i.e., the lowest price it is willing to accept for selling the corresponding resources (Vincent, 1995)) to the auctioneer. 2) **Bid Request Dissemination**: The auctioneer disseminates the received bid request to the relevant DCs whose FL tasks are relevant to the data resources of the DO being auctioned. 3) **Bidding Response**: Each relevant DC evaluates the potential value and cost of the received bid request, and decides on a bid price based on its bidding strategy. The DCs submit their bids to the auctioneer. When

a DC has exhausted its budget, it will forfeit future auctions. 4) **Outcome Determination**: Upon receiving bids from relevant DCs, the auctioneer determines the winning price based on an auction mechanism. It then compares the winning price with the reserve price set by each DO. If the winning price is lower than the reserve price, the auctioneer terminates the auction and informs the DO to initiate another auction for the same resources. Otherwise, the auctioneer informs the winning DC about the cost (i.e., the winning price) it needs to pay, informs the losing DCs, and informs the DO about the winning DC it shall join.

When the auctioning process for session $s$ has been completed or the DC has exhausted its budget, it initiates FL model training with the recruited DOs. Each DC pays the corresponding market prices to the DOs it has recruited.

**FL with Recruited DOs**: After the auction-based DO recruitment process, the DC triggers the FL training process with the recruited DOs in session $s$, which is detailed in Appendix A.1.

Let $v_s^i$ denote the reputation of DO $i \in [C_s]$ (Shi & Yu, 2023) and $x_s^i \in \{0, 1\}$ denote whether the target DC wins $i$. Then, the goal of the target DC across $S$ sessions is to maximize the total utility of winning DOs[2] under the budget $B$, which can be formulated as:

$$\max \sum_{s \in [S]} \sum_{i \in [C_s]} x_s^i \times v_s^i, \quad s.t. \quad \sum_{s \in [S]} \sum_{i \in [C_s]} x_s^i \times p_s^i \le B, \tag{1}$$

**Data Owner Reputation Calculation**: Following (Shi & Yu, 2023; Tang & Yu, 2024b), we calculate the reputation of each DO based on the GTG-Shapley method (Liu et al., 2022) technique and Beta Reputation System (BRS) (Josang & Ismail, 2002).

We start by adopting the SV approach to calculate the contribution $\phi_i$ of each DO $i$ during each training round towards the performance of the resulting FL model as

$$\phi_i = \alpha \sum_{\mathcal{S} \subseteq \mathcal{N} \setminus \{i\}} \frac{f(w_{\mathcal{S} \cup \{i\}}) - f(w_{\mathcal{S}})}{\binom{|\mathcal{N}|-1}{|\mathcal{S}|}}. \tag{2}$$

$\alpha$ is a constant. $\mathcal{S}$ represents the subset of DOs drawn from $\mathcal{N}$. $f(w_{\mathcal{S}})$ denotes the performance of the FL model $w$ when trained on data owned by $\mathcal{S}$. The contributions made by the DOs can be divided into two types: 1) positive contribution (i.e., $\phi_i \ge 0$); and 2) negative contribution (i.e., $\phi_i < 0$). We use the variables $pc_i$ and $nc_i$ to record the number of positive contributions and the number of negative contributions made by each DO $i$, respectively.

---

[1]Following (Tang & Yu, 2023b), we assume that DOs arrive and make their bid requests sequentially, one after the other.

[2]Following (Zhang et al., 2021; Tang & Yu, 2023b; Zhang et al., 2022a;b; Tang & Yu, 2023a), maximizing the total utility is equivalent to optimizing the performance of the global FL model obtained by the target DC.

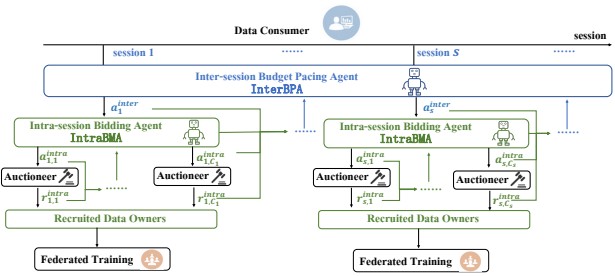

*Figure 2.* An overview of the proposed `MBOS-AFL` approach.

Following BRS, the reputation value $v^i$ of $i$ can be computed as follows:

$$v^i = \mathbb{E}[Beta(pc_i + 1, nc_i + 1)] = \frac{pc_i + 1}{pc_i + nc_i + 2}. \quad (3)$$

It is important to highlight that, as depicted in Eq. (3), the reputation of each DO $i$ undergoes dynamic updates as the FL model training process unfolds. Furthermore, in cases where there is no prior information available, the default initialization for the reputation value of $i$ is set to the uniform distribution, denoted as $v^i = N(0, 1) = Beta(1, 1)$.

The basics of Reinforcement Learning (RL) could be found in Appendix A.2.

## 4. The Proposed `MBOS-AFL` Approach

Our primary objective is to help DCs recruit DOs across multiple sessions while adhering to budget constraints, with the overarching goal of maximizing the total utility. To accomplish this, we must tackle two fundamental challenges: 1) **Budget Allocation**: Determining the allocation of the total budget $B$ to a given session $s$, $B_s$; 2) **Bidding Strategy**: Determining the bid price $b_s^i$ for any given DO $i$ in session $s$ under the session budget $B_s$. Since the AFL market is highly dynamic, it is difficult for DCs to obtain a closed-form analytical solution for the above two problems. Therefore, we design `MBOS-AFL` based on RL (Sutton & Barto, 2018) to solve these problems without requiring prior knowledge. To determine the optimal budget allocation strategy and bidding strategy for a DC to realize the objective outlined in Eq. (1), we design `MBOS-AFL` based on HRL (Pateria et al., 2021). It consists of two HRL-based budget allocation agents: 1) Inter-session Budget Pacing Agent (`InterBPA`), and 2) Intra-session Bidding Agent (`IntraBMA`). An overview of `MBOS-AFL` is shown in Figure 2.

During each FL training session $s$, the `InterBPA` observes the current state within the model training environment. Subsequently, this observed state is channeled into the policy network of the `InterBPA`, generating the recommended inter-session action (i.e., setting the budget $B_s$ for session $s$). This action aims to enhance the current FL model per-

formance, ultimately influencing the outcome across all training sessions. Moreover, this inter-session action serves as an initial state for the `IntraBMA`. It is worth noting that the `InterBPA` will stay static throughout a given session $s$. It is only updated when the session $s$ is concluded. Funneling the inter-session action $B_s$ into the policy network of the `IntraBMA` helps determine the intra-session actions, especially the initial intra-session action.

The primary function of the `IntraBMA` is to help a DC bid for each DO $i \in [C_s]$ in session $s$ in an efficient way, thus contributing to the crafting of the optimal budget allocation strategies under `MBOS-AFL`. The `IntraBMA` takes the dynamic DC state as the input, and produces the optimal action $a_s^i$ as the bid price for data owner $i$ to be submitted to the auctioneer. As a result, the `IntraBMA` will be updated upon every DO auction in session $s$. The synthesis of inter-session and intra-session actions culminates in the formulation of the DC's budget allocation strategy. In the following sections, we provide detailed descriptions of these two agents.

### 4.1. Inter-session Budget Pacing Agent (`InterBPA`)

**State**: The state of the `InterBPA` in session $s \in [S]$, denoted as $s_s^{inter}$, comprises two main segments. The first segment contains historical data derived from the preceding $S'$ sessions. These include the budgets allocated for each of the historical sessions, and the bidding outcomes of `IntraBMA` in these sessions (including the bid prices for DOs, payment for DOs, and reputation of the recruited DOs). The second segment contains current session information (including the number of available DOs and the remaining budget). Thus, the formulation of $s_s^{inter}$ is as follows:

$$s_s^{inter} = \{b_{s-S'}, \cdots, b_{s-1}, p_{s-S'}, \cdots, p_{s-1}, v_{s-S'}, \cdots, v_{s-1}, \\ C_s, B, s\}. \quad (4)$$

$b_{s-1} = \{b_{s-1}^i\}_{t \in [C_{s-1}]}$, $p_{s-1} = \{p_{s-1}^i\}_{i \in [C_{s-1}]}$, and $v_{s-1} = \{v_{s-1}^i\}_{i \in [C_{s-1}]}$. The integration of historical context into the state design is pivotal, as it empowers the agent to understand the impact of its strategies on FL training over time.

**Action**: In session $s$, the action to be taken by the `InterBPA` is to determine the budget allocated to the current session, $a_s^{inter}$, which is expressed as:

$$a_s^{inter} = B_s. \quad (5)$$

Here, $B_s$ denotes the budget for session $s$ to bid for the data owners involved. This inter-session action plays a pivotal role in regulating the amount of budget to be disbursed by the DC during session $s$, thereby helping preserve the total budget $B$ for potential future FL training sessions.

**Reward**: The inter-session reward for session $s$, $r_s^{inter}$, is

determined by the average reputation of DOs recruited in $s$:

$$r_s^{inter} = \frac{1}{\sum_{i \in [C_s]} x_s^i} \sum_{i \in [C_s]} x_s^i v_s^i. \qquad (6)$$

$x_s^i \in \{0, 1\}$ denotes if the DC wins the auction for DO $i$.

**Discount factor**: As the goal of a DC is to maximize the total utility derived from the recruited DOs for a given total budget $B$ regardless of time, the reward discount factor of `InterBPA` is set to 1.

### 4.2. Intra-session Budget Management Agent (`IntraBMA`)

**State**: The state of the `IntraBMA` in session $s$ during an auction for DO $i$, denoted as $s_{s,i}^{intra}$, consists of: 1) $C_s - i$: the remaining DOs in session $s$, 2) $B_s$: the remaining budget of session $s$, and 3) $v_s^i$: the reputation of DO $i$:

$$\boldsymbol{s}_{s,i}^{intra} = \{C_s - i, B_s, v_s^i\}. \qquad (7)$$

**Action**: The action, denoted as $\boldsymbol{a}_{s,i}^{intra}$, to be taken by the `IntraBMA` in session $s$ for DO $i \in [C_s]$ is to determine the bid price for $i$, i.e., $b_s^i$.

**Reward**: The intra-session reward for session $s$ following the bid for DO $i$ is defined as the utility obtained from $i$, which is formulated as:

$$r_{s,i}^{intra} = x_s^i v_s^i. \qquad (8)$$

**Discount factor**: Similar to `InterBPA`, the discount factor for the `IntraBMA` is also set to 1.

### 4.3. Training Procedure for `InterBPA` and `IntraBMA`

Following (Tang & Yu, 2023a), `InterBPA` and `IntraBMA` leverage the Deep Q-Network (DQN) technique (Mnih et al., 2015). Both agents use deep neural networks (DNNs) to model the action-value function $Q(s, a)$, parameterized by $\theta^{inter}$ and $\theta^{intra}$, respectively. To improve stability during training, we pair these networks with a similar DNN architecture parameterized by $\hat{\theta}^{inter}$ and $\hat{\theta}^{intra}$, respectively (referred to as the *target networks*), which also approximates $Q(s, a)$. To update $\theta^{inter}$ and $\theta^{intra}$, the training is conducted by minimizing the following loss function: $\mathcal{L}(\theta) = \frac{1}{2}\mathbb{E}_{(s,a,r,s') \sim \mathcal{D}}[(y - Q(s, a; \theta))^2]$. The *replay buffer*, $\mathcal{D}$, is a storage mechanism for transition tuples $\{(s, a, r, s')\}_{i=1}^n$, where $s'$ is the new observation following action $a$ based on the state $s$, resulting in reward $r$. This buffer allows the agent to learn from its past experiences by randomly sampling batches of transitions during training. $y$ represents the temporal difference target, and is computed as $y = r + \gamma \max_{a'} Q(s, a'; \hat{\theta})$. $\gamma$ is the discount factor, $\hat{\theta}$ represents the parameters of the target network

---

**Algorithm 1** The training procedure of `MBOS-AFL`

Initialize $Q^{intra}$, $Q^{inter}$ with parameters $\theta^{intra}$, $\theta^{inter}$; target networks of $Q^{intra}$ and $Q^{inter}$ with parameters $\hat{\theta}^{intra}$ and $\hat{\theta}^{inter}$; replay memories $\mathcal{D}^{intra}$ and $\mathcal{D}^{inter}$; target networks' update frequency $\Gamma$.

1: **for** $s \in [S]$ **do**
2:     Observe state $s_s^{inter}$;
3:     Compute $B_s$ according to $\epsilon$-greedy policy w.r.t $Q^{inter}$;
4:     **for** $i \in [C_s]$ **do**
5:         Observe state $s_{s,i}^{intra}$;
6:         Compute $b_s^i$ according to $\epsilon$-greedy policy w.r.t $Q^{intra}$;
7:         Submit $b_s^i$ to the auctioneer;
8:         Obtain rewards $v_s^i$ and the payment $p_s^i$;
9:         $B_s \leftarrow B_s - p_s^i$;
10:      Store transition tuples in $\mathcal{D}^{intra}$;
11:      Sample a random minibatch of $m$ samples from $\mathcal{D}$;
12:      $y^{intra} = r_s^i + \gamma \max_{a_s^{intra'}} Q^{intra}(s_{s,i+1}^{intra}, a_s^{intra'}; \hat{\theta}^{intra})$;
13:      Update $\theta^{intra}$ by minimizing $\sum_m [(y^{intra} - Q^{intra}(s_{s,i}^{intra}, a_{s,i}^{intra}; \theta^{intra})^2]$;
14:      $\hat{\theta}^{intra} \leftarrow \theta^{intra}$ every $\Gamma$ steps;
15:     **end for**
16:     Obtain rewards $r_s^{inter}$ and the total payment $p_s^i$ during session $s$;
17:     $B \leftarrow B - \sum_{i \in [C_s]} p_s^i$;
18:     Store transition tuples in $\mathcal{D}^{inter}$;
19:     Sample a random minibatch of $m$ samples from $\mathcal{D}$;
20:     $y^{inter} = r_s + \gamma \max_{a_s^{inter'}} Q^{inter}(s_{s+1}^{inter}, a_s^{inter'}; \hat{\theta}^{inter})$;
21:     Update $\theta^{inter}$ by minimizing $\sum_m [(y^{inter} - Q^{inter}(s_s^{inter}, a_s^{inter}; \theta^{inter})^2]$;
22:     $\hat{\theta}^{inter} \leftarrow \theta^{inter}$ every $\Gamma$ steps;
23: **end for**

---

associated with the corresponding agent. $Q(s, a'; \hat{\theta})$ is the predicted action-value function of the corresponding agent for its next state $s'$ and all possible actions $a'$. This target network is used to stabilize the learning process by providing a fixed target during training, which is updated periodically (every $\Gamma$ steps) to match the current action-value network. Algorithm 1 is the training procedure for `MBOS-AFL`.

## 5. Experimental Evaluation

### 5.1. Experiment Settings

**Dataset**: The performance assessment of `MBOS-AFL` is conducted on the following six widely-adopted datasets in FL studies: 1) MNIST[3], 2) CIFAR-10[4], 3) Fashion-MNIST (i.e., FMNIST) (Xiao et al., 2017), 4) EMNIST-digits (i.e., EMNISTD), 5) EMNIST-letters (i.e., EMNISTL) (Cohen et al., 2017) and 6) Kuzushiji-MNIST (i.e., KM-NIST) (Clanuwat et al., 2018). The FL models used are the same as those employed in (Tang & Yu, 2023b).

---

[3]http://yann.lecun.com/exdb/mnist/
[4]https://www.cs.toronto.edu/kriz/cifar.html

**Comparison Approaches**: We evaluate the performance of MBOS-AFL against the following seven AFL bidding approaches in our experiments: Constant Bid (**Const**) (Zhang et al., 2014), Randomly Generated Bid (**Rand**) (Zhang et al., 2021; 2022b), Below Max Utility Bid (**Bmub**), Linear-Form Bid (**Lin**) (Perlich et al., 2012), Bidding Machine (**BM**) (Ren et al., 2017), Reinforcement Learning-based Bid (**RLB**) (Tang & Yu, 2023a; 2024b), FedBidder-sim (**FBs**), and Fed-Bidder-com (**FBc**) (Tang & Yu, 2023b). Details can be found in (Tang et al., 2024b).

**Experiment Scenarios**: We compare MBOS-AFL with baselines under two main experiment scenarios with each containing 10,000 DOs: 1) **IID data, varying dataset sizes, without noise**: In this scenario, the sizes of datasets owned by various DOs are randomly generated, ranging from 500 to 5,000 samples. Additionally, all the data are independent and identically distributed (IID), with no noise. 2) **Non-IID data, with noise**: In this experimental scenario, we deliberately introduce data heterogeneity by adjusting the class distribution among individual DOs. Following (Tang et al., 2024b), we implement the following Non-IID setup. We designate 1 class (on datasets other than EMNISTL) or 6 classes (on EMNISTL) as the minority class and assign this minority class to 100 DOs. As a result, these 100 DOs possess images for all classes, while all other DOs exclusively have images for the remaining nine classes, excluding the minority class. In this experiment scenario, each DO holds 3,000 images. Additionally, we simulate scenarios in which the minority DOs contain 10% or 25% noisy data. The implementation details can be found in Appendix A.3.

**Evaluation Metrics**: To evaluate the effectiveness of all the comparison methods, we adopt the following three metrics: 1) the number of data samples won by the DC (**#data**), 2) the utility obtained by the DC (**utility**), and 3) the test accuracy (**Acc**). More details could be found in Appendix A.4.

### 5.2. Results and Discussion

To conduct a comparative analysis of bidding strategies based on these metrics, we carry out experiments across six datasets, each with varying budget settings. These settings span the range of $\{100, 200, 400, 600, 800\}$. The results are shown in Tables 1, 2, and Figure 3.

Table 1 shows the results of various comparison methods under the IID data, different sizes of DOs datasets without noisy samples scenario. It can be observed that under all six datasets and five budget settings, MBOS-AFL consistently outperforms all baseline methods in terms of both evaluation metrics. Specifically, compared to the best-performing baseline, MBOS-AFL achieves 12.28% and 14.52% improvement in terms of total utility and the number of data samples won, respectively. Figure 3 shows the corresponding test accuracy. The results align with the auction performance

shown in Table 1 with MBOS-AFL improving the test accuracy by 1.23% on average.

In addition, the comparative results under the Non-IID data with noise scenario can be found in Table 2. It can be observed that under these two different settings, the proposed method MBOS-AFL consistently outperforms existing methods in terms of achieving higher FL model accuracy. In particular, on average, MBOS-AFL achieves 1.49% and 1.72% higher FL model accuracy compared to the best performance achieved by baselines under the 10% noisy data and 25% noisy data settings, respectively. All these results demonstrate the effectiveness of our approach in helping DCs optimize their budget pacing and bidding strategies for DOs under the emerging multi-session AFL scenarios.

Lin and Bmub typically outperform Const and Rand due to the use of utility in the bidding process. However, Bmub is less effective than Lin due to the reliance on randomness. Meanwhile, the more advanced methods BM, FBs, FBc, RLB and MBOS-AFL perform significantly better than the simpler approaches. This is largely due to the inclusion of auction records (including auction history and bidding records) and the use of advanced learning methods.

RLB and MBOS-AFL both outperform BM, FBs, and FBc, due to their ability of adaptive adjustment to the highly dynamic auction environment. While BM does consider market price distribution, it derives this distribution by learning the prediction of each bid request's market price density, which may lead to overfitting. In contrast, FBs and FBc obtain the market price distribution via a predefined winning function, which helps predict the expected bid costs more accurately. However, BM, FBs and FB are still static bidding strategies. They are essentially represented by linear or non-linear functions whose parameters are derived from historical auction data using heuristic techniques. Subsequently, these parameters are applied to new auctions, even if the dynamics of these new auctions may vary significantly from those in the historical data. The inherent dynamism of the AFL market poses a considerable challenge for these static bidding methods, making it hard for them to consistently achieve desired outcomes in subsequent auctions.

While RLB optimize its bidding process with dynamic programming, it is susceptible to the drawback of immediate rewards, which might result in indiscriminate bidding for data samples without considering their associated costs. This issue is effectively addressed by MBOS-AFL. Moreover, it is worth highlighting that RLB is not designed for optimizing budget allocation across multiple sessions. This is a distinction where MBOS-AFL offers significant advantages.

The test accuracy achieved by the FL models trained under all bidding strategies on CIFAR-10 is consistently lower than that on other datasets. This can be attributed to the base

*Table 1.* Comparison results under the scenario of IID data, different sizes of DOs datasets without noisy samples. The best results are highlighted in **Bold**. *Ours* represents `MBOS-AFL`.

| Budget | Method | MNIST | | CIFAR | | FMNIST | | EMNIST | | EMNISTL | | KMNIST | |
|---|---|---|---|---|---|---|---|---|---|---|---|---|---|
| | | #data | utility | #data | utility | #data | utility | #data | utility | #data | utility | #data | utility |
| 100 | Const | 8,832 | 7.36 | 9,897 | 7.87 | 10,722 | 6.46 | 7,638 | 6.52 | 7,359 | 7.02 | 7,810 | 6.75 |
| | Rand | 9,125 | 8.41 | 8,721 | 8.43 | 9,743 | 8.09 | 8,853 | 8.10 | 6,822 | 7.97 | 8,940 | 7.96 |
| | Bmub | 9,246 | 9.03 | 11,302 | 9.19 | 12,274 | 8.76 | 10,382 | 8.91 | 6,485 | 9.15 | 10,551 | 8.62 |
| | Lin | 9,461 | 10.28 | 11,426 | 10.17 | 13,523 | 9.84 | 10,673 | 10.33 | 8,220 | 10.51 | 10,694 | 9.97 |
| | BM | 12,324 | 11.95 | 13,367 | 11.85 | 15,321 | 12.65 | 14,399 | 12.19 | 15,157 | 12.27 | 14,501 | 12.46 |
| | FBs | 13,985 | 14.51 | 14,259 | 13.51 | 16,373 | 13.53 | 15,321 | 13.46 | 14,408 | 13.44 | 15,509 | 13.54 |
| | FBc | 13,869 | 13.84 | 13,984 | 13.70 | 15,843 | 13.42 | 16,772 | 14.23 | 14,168 | 13.67 | 16,927 | 13.64 |
| | RLB | 13,892 | 14.42 | 14,263 | 14.26 | 17,783 | 13.95 | 15,989 | 13.51 | 15,544 | 14.40 | 16,027 | 14.33 |
| | Ours | **14,944** | **16.59** | **17,397** | **17.47** | **19,064** | **18.19** | **18,674** | **17.46** | **16,317** | **18.59** | **18,687** | **16.55** |
| 200 | Const | 11,037 | 8.49 | 12,043 | 9.31 | 16,374 | 8.52 | 13,826 | 9.46 | 10,876 | 10.33 | 13,950 | 9.31 |
| | Rand | 10,895 | 10.06 | 11,894 | 10.00 | 14,898 | 9.90 | 12,452 | 10.34 | 12,808 | 10.42 | 12,601 | 10.05 |
| | Bmub | 16,582 | 9.58 | 17,021 | 10.60 | 25,327 | 10.60 | 17,817 | 10.40 | 20,966 | 11.43 | 17,878 | 10.97 |
| | Lin | 17,803 | 13.14 | 17,849 | 12.88 | 26,880 | 12.88 | 19,435 | 12.64 | 27,860 | 12.70 | 19,553 | 12.97 |
| | BM | 23,584 | 14.97 | 20,836 | 15.11 | 31,945 | 15.92 | 21,656 | 15.03 | 35,016 | 15.29 | 21,722 | 15.70 |
| | FBs | 27,813 | 17.70 | 28,456 | 17.61 | 34,936 | 17.09 | 26,994 | 17.01 | 31,743 | 17.40 | 27,087 | 17.49 |
| | FBc | 28,005 | 17.51 | 29,835 | 17.24 | 36,873 | 17.58 | 27,863 | 16.60 | 34,686 | 16.99 | 27,892 | 17.89 |
| | RLB | 29,468 | 17.77 | 30,138 | 17.82 | 35,548 | 17.04 | 26,748 | 17.45 | 37,122 | 17.82 | 26,819 | 17.23 |
| | Ours | **33,045** | **21.99** | **35,163** | **21.08** | **39,982** | **23.72** | **35,656** | **19.59** | **37,645** | **22.43** | **35,737** | **18.08** |
| 400 | Const | 14,395 | 8.72 | 15,362 | 8.11 | 18,475 | 8.34 | 17,877 | 7.82 | 10,177 | 8.04 | 17,940 | 8.41 |
| | Rand | 13,195 | 9.86 | 16,372 | 9.71 | 17,844 | 6.87 | 17,003 | 7.13 | 6,431 | 9.02 | 17,051 | 9.20 |
| | Bmub | 23,378 | 10.90 | 25,631 | 11.16 | 31,487 | 10.86 | 24,756 | 10.05 | 23,639 | 10.63 | 24,869 | 11.33 |
| | Lin | 24,523 | 14.58 | 26,830 | 14.41 | 32,677 | 14.24 | 25,669 | 14.28 | 36,261 | 14.31 | 25,802 | 14.46 |
| | BM | 38,516 | 16.46 | 30,173 | 16.54 | 38,552 | 16.90 | 30,878 | 17.26 | 41,050 | 17.66 | 31,077 | 17.61 |
| | FBs | 50,983 | 19.32 | 38,452 | 19.24 | 39,236 | 18.54 | 38,452 | 18.69 | 40,605 | 19.04 | 38,566 | 19.09 |
| | FBc | 50,146 | 19.23 | 39,817 | 19.10 | 41,582 | 18.37 | 40,663 | 18.40 | 39,555 | 18.85 | 40,768 | 18.88 |
| | RLB | 51,643 | 19.54 | 42,731 | 19.63 | 45,667 | 18.84 | 37,748 | 19.18 | 43,077 | 19.71 | 37,843 | 19.55 |
| | Ours | **56,872** | **23.65** | **53,672** | **22.71** | **52,386** | **23.00** | **47,135** | **19.32** | **46,341** | **23.83** | **47,262** | **19.73** |
| 600 | Const | 17,895 | 9.71 | 19,378 | 9.60 | 21,394 | 9.33 | 19,832 | 10.08 | 10,596 | 9.55 | 19,982 | 8.92 |
| | Rand | 19,803 | 8.68 | 20,184 | 9.07 | 20,853 | 11.69 | 18,838 | 10.37 | 24,581 | 9.15 | 18,966 | 9.83 |
| | Bmub | 30,164 | 12.07 | 29,174 | 11.93 | 37,421 | 11.85 | 29,669 | 12.06 | 33,768 | 11.94 | 29,845 | 11.97 |
| | Lin | 32,973 | 15.62 | 30,375 | 15.59 | 40,128 | 15.08 | 34,452 | 15.16 | 47,484 | 15.61 | 34,629 | 15.62 |
| | BM | 49,807 | 17.09 | 49,272 | 17.43 | 47,533 | 18.06 | 38,743 | 17.85 | 51,454 | 18.23 | 38,943 | 18.54 |
| | FBs | 62,396 | 20.49 | 50,384 | 20.58 | 46,731 | 19.54 | 45,232 | 19.64 | 50,482 | 20.29 | 45,288 | 20.29 |
| | FBc | 61,478 | 20.31 | 52,836 | 20.24 | 52,843 | 19.92 | 48,767 | 19.38 | 49,468 | 20.04 | 48,958 | 20.06 |
| | RLB | 63,672 | 20.64 | 58,273 | 20.64 | 50,472 | 19.26 | 42,534 | 19.69 | 59,455 | 20.53 | 42,692 | 20.44 |
| | Ours | **66,654** | **21.72** | **60,737** | **22.82** | **63,824** | **24.17** | **58,462** | **23.01** | **63,441** | **23.54** | **58,522** | **21.72** |
| 800 | Const | 23,047 | 11.04 | 24,753 | 11.35 | 26,311 | 11.13 | 22,644 | 10.79 | 17,875 | 11.40 | 22,705 | 11.30 |
| | Rand | 24,853 | 14.09 | 22,845 | 13.34 | 22,734 | 13.68 | 20,474 | 13.60 | 26,563 | 13.57 | 20,642 | 13.26 |
| | Bmub | 36,703 | 12.99 | 35,777 | 12.70 | 40,275 | 13.47 | 36,648 | 12.91 | 38,570 | 13.08 | 36,732 | 13.17 |
| | Lin | 39,651 | 16.79 | 38,561 | 16.88 | 47,823 | 16.55 | 40,537 | 16.67 | 59,390 | 16.86 | 40,727 | 16.76 |
| | BM | 57,442 | 18.57 | 52,735 | 18.68 | 51,272 | 19.16 | 46,772 | 19.34 | 65,086 | 19.41 | 46,933 | 19.59 |
| | FBs | 70,496 | 22.09 | 62,842 | 22.07 | 54,453 | 21.07 | 51,863 | 21.02 | 67,470 | 21.54 | 51,942 | 21.69 |
| | FBc | 72,845 | 22.04 | 63,112 | 22.06 | 55,388 | 21.18 | 56,991 | 21.09 | 61,598 | 21.57 | 57,152 | 21.53 |
| | RLB | 70,381 | 22.31 | 66,843 | 22.37 | 52,621 | 20.92 | 53,823 | 20.95 | 68,943 | 21.78 | 57,900 | 21.92 |
| | Ours | **77,821** | **22.40** | **71,244** | **23.46** | **64,739** | **23.12** | **62,579** | **22.57** | **70,393** | **23.04** | **59,711** | **22.18** |

*Figure 3.* Comparison of accuracy under the scenario of IID data, different sizes of DOs datasets without noisy samples.

*Table 2.* Comparison of accuracy under the Non-IID data with noise scenario. *10%* and *25%* represents 10% and 25% noisy data, respectively. Bud. represent budget and *Ours* represents `MBOS-AFL`.

| Bud. | Method | MNIST | | CIFAR | | FMNIST | | EMNIST | | EMNISTL | | KMNIST | |
|---|---|---|---|---|---|---|---|---|---|---|---|---|---|
| | | 10% | 25% | 10% | 25% | 10% | 25% | 10% | 25% | 10% | 25% | 10% | 25% |
| 100 | Const | 70.11 | 70.03 | 12.88 | 13.97 | 61.48 | 57.87 | 77.02 | 76.46 | 64.92 | 63.30 | 58.21 | 59.63 |
| | Rand | 69.61 | 65.42 | 10.57 | 10.83 | 62.70 | 59.48 | 78.69 | 77.97 | 63.97 | 62.83 | 57.01 | 59.12 |
| | Bmub | 71.22 | 70.61 | 15.37 | 12.94 | 63.32 | 60.45 | 78.42 | 77.37 | 66.88 | 65.19 | 61.83 | 61.76 |
| | Lin | 72.36 | 70.32 | 18.65 | 17.41 | 64.04 | 64.13 | 78.62 | 77.44 | 66.47 | 64.07 | 62.72 | 62.97 |
| | BM | 72.31 | 71.65 | 19.50 | 19.62 | 67.35 | 66.25 | 79.50 | 78.42 | 67.17 | 64.62 | 64.55 | 63.77 |
| | FBs | 73.23 | 72.32 | 23.59 | 22.03 | 70.97 | 70.26 | 79.51 | 78.35 | 68.35 | 65.94 | 65.82 | 64.33 |
| | FBc | 73.11 | 74.80 | 23.42 | 22.26 | 71.29 | 70.68 | 79.92 | 78.93 | 67.69 | 64.78 | 65.47 | 63.88 |
| | RLB | 73.07 | 73.11 | 22.94 | 22.98 | 71.03 | 69.55 | 79.83 | 78.66 | 68.20 | 65.57 | 65.38 | 63.93 |
| | *Ours* | **73.79** | **75.22** | **23.88** | **23.24** | **72.31** | **71.42** | **80.66** | **79.29** | **69.26** | **66.76** | **66.15** | **65.08** |
| 200 | Const | 70.73 | 66.38 | 10.68 | 11.08 | 63.74 | 60.16 | 77.98 | 77.52 | 67.84 | 66.16 | 58.44 | 58.29 |
| | Rand | 69.48 | 68.96 | 10.32 | 10.26 | 63.86 | 59.63 | 78.63 | 78.19 | 68.24 | 66.88 | 59.25 | 58.09 |
| | Bmub | 71.81 | 70.52 | 13.39 | 13.03 | 63.83 | 62.18 | 79.37 | 78.37 | 69.09 | 67.42 | 63.04 | 63.34 |
| | Lin | 72.98 | 70.55 | 19.07 | 17.96 | 64.43 | 64.16 | 79.43 | 78.43 | 69.96 | 68.44 | 67.07 | 66.09 |
| | BM | 73.43 | 72.48 | 20.36 | 20.14 | 64.53 | 70.01 | 80.52 | 79.40 | 70.19 | 67.35 | 69.01 | 67.63 |
| | FBs | 74.69 | 72.17 | 23.82 | 22.79 | 71.49 | 71.99 | 80.28 | 79.27 | 69.65 | 67.57 | 68.77 | 68.69 |
| | FBc | 74.29 | 72.99 | 23.61 | 22.58 | 71.86 | 71.61 | 80.37 | 79.52 | 70.70 | 68.45 | 68.75 | 67.04 |
| | RLB | 74.33 | 73.26 | 23.77 | 23.14 | 71.52 | 70.74 | 80.48 | 79.52 | 70.13 | 68.11 | 70.52 | 70.48 |
| | *Ours* | **75.60** | **75.72** | **24.94** | **24.52** | **72.98** | **73.13** | **81.31** | **80.10** | **71.39** | **69.05** | **71.13** | **71.27** |
| 400 | Const | 71.06 | 68.34 | 17.09 | 16.96 | 64.01 | 58.93 | 78.49 | 77.98 | 68.19 | 66.69 | 68.66 | 68.33 |
| | Rand | 70.05 | 67.74 | 20.90 | 20.45 | 64.25 | 60.58 | 78.62 | 78.43 | 68.88 | 67.64 | 70.36 | 69.75 |
| | Bmub | 72.27 | 70.26 | 22.21 | 20.49 | 64.37 | 63.15 | 79.97 | 78.90 | 69.71 | 68.11 | 69.93 | 68.56 |
| | Lin | 72.99 | 71.02 | 24.18 | 22.94 | 65.52 | 65.44 | 80.01 | 78.99 | 70.53 | 69.12 | 70.37 | 69.10 |
| | BM | 74.96 | 73.01 | 25.59 | 23.74 | 65.87 | 68.38 | 80.90 | 79.91 | 71.62 | 70.35 | 71.58 | 70.44 |
| | FBs | 75.85 | 73.53 | 26.47 | 24.50 | 71.72 | 70.06 | 81.36 | 80.22 | 71.75 | 70.17 | 71.93 | 70.85 |
| | FBc | 75.66 | 73.77 | 26.21 | 24.27 | 72.03 | 71.95 | 81.29 | 80.18 | 71.88 | 70.38 | 71.01 | 69.56 |
| | RLB | 75.25 | 74.96 | 26.78 | 24.83 | 72.31 | 72.24 | 81.55 | 80.47 | 71.99 | 70.59 | 72.45 | 70.72 |
| | *Ours* | **76.59** | **76.33** | **27.65** | **25.86** | **73.85** | **73.63** | **81.86** | **80.69** | **72.54** | **71.84** | **73.38** | **71.66** |
| 600 | Const | 71.05 | 69.36 | 23.10 | 21.66 | 64.61 | 61.77 | 79.28 | 78.49 | 68.39 | 67.01 | 69.21 | 68.69 |
| | Rand | 68.79 | 69.05 | 22.72 | 20.32 | 64.39 | 62.49 | 79.25 | 78.83 | 69.31 | 67.95 | 70.19 | 69.74 |
| | Bmub | 71.95 | 71.07 | 18.90 | 22.02 | 64.41 | 63.78 | 80.68 | 79.38 | 70.49 | 68.71 | 70.78 | 69.60 |
| | Lin | 73.54 | 72.57 | 24.43 | 24.79 | 66.92 | 66.18 | 80.86 | 79.58 | 71.44 | 69.92 | 71.21 | 69.94 |
| | BM | 75.25 | 73.58 | 28.30 | 26.62 | 67.21 | 67.80 | 81.42 | 80.26 | 72.47 | 71.07 | 71.97 | 70.82 |
| | FBs | 76.18 | 74.16 | 28.85 | 27.25 | 73.55 | 71.81 | 81.47 | 80.34 | 72.51 | 71.06 | 72.26 | 72.23 |
| | FBc | 76.25 | 73.98 | 29.07 | 28.95 | 74.14 | 73.31 | 81.49 | 80.31 | 72.51 | 70.99 | 72.18 | 72.84 |
| | RLB | 76.06 | 73.15 | 28.52 | 29.60 | 73.85 | 73.05 | 81.68 | 80.60 | 73.07 | 71.64 | 73.41 | 72.81 |
| | *Ours* | **76.93** | **76.71** | **29.91** | **30.55** | **74.46** | **74.05** | **82.16** | **80.93** | **73.21** | **71.86** | **74.63** | **73.79** |
| 800 | Const | 67.21 | 66.43 | 23.63 | 21.95 | 68.17 | 64.97 | 79.64 | 78.81 | 68.85 | 67.49 | 69.49 | 69.01 |
| | Rand | 68.95 | 71.02 | 24.54 | 20.66 | 68.15 | 65.32 | 79.78 | 79.23 | 70.13 | 68.75 | 70.91 | 70.11 |
| | Bmub | 71.90 | 72.16 | 25.97 | 19.45 | 69.24 | 66.51 | 81.08 | 79.77 | 70.80 | 69.05 | 71.52 | 70.60 |
| | Lin | 75.11 | 72.66 | 25.46 | 28.06 | 71.87 | 69.03 | 81.37 | 80.12 | 71.61 | 70.16 | 71.76 | 70.46 |
| | BM | 75.28 | 73.89 | 28.76 | 29.00 | 72.83 | 70.31 | 81.64 | 80.58 | 72.89 | 71.63 | 73.09 | 71.74 |
| | FBs | 76.09 | 75.04 | 29.54 | 30.18 | 75.92 | 73.86 | 81.87 | 80.83 | 72.99 | 71.72 | 73.42 | 72.20 |
| | RLB | 76.31 | 76.34 | 30.05 | 30.81 | 76.39 | 74.72 | 82.06 | 81.07 | 73.62 | 72.37 | 74.90 | 73.18 |
| | *Ours* | **77.29** | **76.78** | **32.82** | **32.46** | **77.10** | **75.57** | **82.47** | **82.69** | **73.77** | **73.55** | **75.39** | **73.82** |

model adopted for FL training. As mentioned in Section 5.1, the accuracy reported in these two figures is with regard to the VGG11 network. Nevertheless, even with such a less effective base model, `MBOS-AFL` still significantly outperforms other baselines.

## 6. Conclusions

In this paper, we propose the Multi-session Budget Optimization Strategy for forward Auction-based FL (`MBOS-AFL`). It is designed to empower FL DCs with the ability to strategically allocate budgets over multiple FL training sessions and judiciously distribute the budget among DOs within each session by bidding with different bid prices, in order to maximize total utility. Based on the hierarchical reinforcement learning, `MBOS-AFL` jointly optimizes inter-session budget pacing and intra-session bidding for DCs in the AFL ecosystem. To the best of our knowledge, it is the first budget optimization decision support method with budget pacing capability designed for DCs in multi-session forward auction-based FL.

## Acknowledgements

This research is supported, in part, by the Ministry of Education, Singapore, under its Academic Research Fund Tier 1 (RG101/24); the RIE2025 Industry Alignment Fund – Industry Collaboration Projects (IAF-ICP) (Award I2301E0026), administered by A*STAR, as well as supported by Alibaba Group and NTU Singapore through Alibaba-NTU Global e-Sustainability CorpLab (ANGEL); and the National Research Foundation, Singapore and DSO National Laboratories under the AI Singapore Programme (AISG Award No. AISG2-RP-2020-019). Xiaoxiao Li is supported by CIFAR AI Chair Awards and Canada Research Chair Fellowship.

## Impact Statement

This paper presents work whose goal is to advance the field of Machine Learning. There are many potential societal consequences of our work, none which we feel must be specifically highlighted here.

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

# A. Appendix

## A.1. Federated Learning with Recruited Data Owners

After the auction-based DO recruitment process, the DC triggers the FL training process with the recruited DOs in session $s$, which is detailed in Appendix A.1. Specifically, the FL process operates through communication between the recruited DOs and the target DC in a round-by-round manner. In each training round $t$ in session $s$, the target DC broadcasts the current global model parameters $\boldsymbol{w}_s^{t-1}$ to the recruited DOs. Upon receiving $\boldsymbol{w}_s^{t-1}$, each DO $i$ performs a local update to obtain $\boldsymbol{w}_{s,i}^t$ based on its private data $D_i$, guided by the objective function

$$\underset{\boldsymbol{w}_{s,i}^t}{\arg\min} \, \mathbb{E}_{(\boldsymbol{x},y)\sim D_i}[\mathcal{L}(\boldsymbol{w}_{s,i}^t; (\boldsymbol{x}, y)]. \tag{9}$$

$\mathcal{L}(\cdot)$ represents the loss function, which depends on the FL model aggregation algorithm and the current global model parameters $\boldsymbol{w}_s^{t-1}$. For instance, FedAvg (McMahan et al., 2017) calculates $\boldsymbol{w}_{s,i}^t$ by employing SGD (Robbins & Monro, 1951) for a certain number of epochs using the cross-entropy loss. At the end of round $t$, DO $i$ sends its optimized parameters $\boldsymbol{w}_{s,i}^t$ to the target DC. The global model is then updated by aggregating these parameter updates from the DOs as

$$\boldsymbol{w}_s^t = \sum_i \frac{|D_i|}{\sum_i |D_i|} \boldsymbol{w}_{s,i}^t. \tag{10}$$

$\sum_i |D_i|$ denotes the total number of data samples of all the recruited DOs in session $s$.

## A.2. Reinforcement Learning Basics

A Markov Decision Process (MDP) is a mathematical framework for modeling decision-making in which an agent interacts with an environment through discrete time steps. MDP is formally defined by the tuple $\langle S, A, P, R, \gamma \rangle$: 1) $S$ represents the possible states in the environment, denoted as $s \in S$. 2) $A$ encompasses the feasible actions the agent can take. 3) $P : S \times A \times S \rightarrow [0, 1]$ is the transition probability function for the likelihood of transitioning between states when an action is taken, capturing environmental dynamics. 4) $R : S \times A \times S \rightarrow \mathbb{R}$ is the reward function, specifying immediate rewards upon state transitions due to specific actions, with the agent's aim to maximize cumulative rewards. 5) $\gamma \in [0, 1]$ serves as the discount factor, reflecting the agent's preference for immediate rewards versus future rewards.

During the MDP process, the agent interacts with the environment across discrete time steps. At each time step, it selects an action $a \in A$ based on policy $\pi : S \rightarrow A$, subsequently receiving a reward $r$, and the environment undergoes state transitions according to $P$.

The goal of MDP is to identify an optimal policy $\pi : S \rightarrow A$ that maximizes the expected sum of discounted rewards over time, given by $\max_\pi \mathbb{E}\left[\sum_{t=1}^T \gamma^{t-1} r^t\right]$. This entails finding the policy maximizing expected cumulative rewards. The value function $V^\pi : S \rightarrow \mathbb{R}$ is associated with each policy, quantifying expected cumulative rewards. The optimal value function $V^* : S \rightarrow \mathbb{R}$ represents the maximum achievable expected cumulative reward achievable with the best policy from each state.

## A.3. Implementation Details

In our experiments, we faced the challenge of not having a publicly available AFL bidding behaviour dataset. To address this issue, we track the behaviors of DCs over time during simulations to gradually accumulate data in four different settings. Each setting contains 160 DCs who adopted one of the eight bidding strategies listed in the Compared Approaches section.

In the first setting, each of the eight baseline bidding methods is adopted by one eighth of the DCs. In the second setting, as BM, Fed-Bidder variants (FBs and FBc) and RLB have AI techniques similar to `MBOS-AFL`, these four bidding strategies are adopted by three sixteenths of the total population, while the remaining four baselines are adopted by one sixteenth of the total population. In the third and fourth settings, as both Fed-Bidder variants and `MBOS-AFL` are designed specifically for AFL, we set the percentage of DCs adopting FBs and FBc to be higher than those adopting the other six baselines. Specifically, under the third setting, 50 DCs adopt FBs and FBc, while 10 DCs adopt each of the other six baselines. Under the fourth setting, 65 DCs adopted FBs and FBc, while 5 DCs adopted each of the other six baselines. We adopt the second-price sealed-bid (SPSB) auction mechanism in our experiments. By tracking the behaviors of DCs over time, we can gradually accumulate data in the absence of a publicly available dataset related to AFL bidding behaviours.

To evaluate the effectiveness of `MBOS-AFL`, we create nine DCs, each utilizing one of the aforementioned bidding approaches to join the auction for each bid request (i.e., each DO) in each session $s$. Following (Tang & Yu, 2023b), bid requests are delivered in chronological order. Upon receiving a bid request, each DC derives its bid price based on its adopted bidding strategy. Subsequently, the auctioneer gathers the bid prices, identifies the winner, and determines the market price using the SPSB auction mechanism. The winning DC pays the market price to the DO. The process concludes when there are no more bid requests or when the budget is depleted.

`MBOS-AFL` utilizes fully connected neural networks with three hidden layers each containing 64 nodes to generate bid prices for a target DO on behalf of their respective DCs. The replay buffer $\mathcal{D}$ of both the `InterBPA` and the `IntraBMA` are set to 5,000. During training, both agents explore the environment using an $\epsilon$-greedy policy with an annealing rate from 1.0 to 0.05. In updating both $Q^{intra}$ and $Q^{inter}$, 64 tuples uniformly sampled from $\mathcal{D}$ are used for each training step, and the corresponding target networks are updated once every 20 steps. In our experiments, we use RMSprop with a learning rate of 0.0005 to train all neural networks, and set the discount factor $\gamma$ to 1. In addition, we have set the number of candidate DOs within each session to 200 (i.e., $C_s = 200$). The communication round in each session is set at 100, while the local training epoch is set at 30. All experiments were conducted five times, and the averaged results are reported.

### A.4. Evaluation Metrics

To evaluate the effectiveness of all the comparison methods, we adopt the following three metrics: 1) The number of data samples won by the DC (**#data**) is defined as the cumulative number of data samples owned by all DOs recruited by the corresponding DC until the budget or session limits are reached. 2) The utility obtained by the DC (**utility**) is defined as the cumulative reputation of DOs recruited by the corresponding DC until the budget or session limits are reached. 3) The test accuracy (**Acc**) is determined as the accuracy of the final FL model for the respective DC, up to the point where either the budget or session limits are reached.

