# OpenReview forum: "Multi-Session Budget Optimization for Forward Auction-based Federated Learning"
_ICML.cc/2025/Conference — ICML 2025 poster_

### Official Review · Reviewer_KgPU · 2025-03-03

**Overall Recommendation:** 4

**Summary:**

This paper explores the design of bidding strategies for data consumers in auction-based federated learning (AFL), involving three key stakeholders: (1) data owners, who are willing to share their potentially sensitive data in exchange for appropriate compensation; (2) data consumers, who require data to train their federated learning models; and (3) a trusted third-party “auctioneer,” responsible for orchestrating auctions between data owners and consumers to optimize the latter’s key performance metrics. typically assume a single-session setting. However, in practice, data consumers often recruit data owners across multiple training sessions. This limitation in existing works leads to suboptimal performance, as they fail to account for budget allocation across sessions and the complex interplay between inter-session and intra-session dynamics. This paper presents a novel budget allocation strategy under multi-session FL sessions for data consumers in AFL. By tackling a critical issue for AFL sustainability through a theoretically grounded and empirically validated approach, this work makes a valuable contribution to the field

**Claims And Evidence:**

Extensive experiments across diverse scenarios (IID/Non-IID data, noisy settings, varying budgets) strongly support the claims. The results hold consistently across key metrics (utility, data volume, accuracy), and comparisons with baselines (e.g., RLB, Fed-Bidder) confirm the advantage of hierarchical RL. Reproducibility is ensured with detailed implementation guidelines in the appendix.

**Essential References Not Discussed:**

No critical omissions found.

**Experimental Designs Or Analyses:**

The experiments conducted are rigorous: five runs with averaged results, varied budgets, and multiple noise levels. In addition, the inclusion of both IID and Non-IID settings strengthens validity.

**Methods And Evaluation Criteria:**

The hierarchical RL framework well suit the inter-session budget allocation and intra-session budget allocation case, addressing both pacing and bidding dynamics. The datasets used in this paper are widely adopted in the filed of AFL and considering of realistic scenarios (e.g., noisy, Non-IID data) ensures thorough evaluation. Metrics (utility, accuracy) align with the DC’s goal of maximizing model performance under budget constraints

**Other Comments Or Suggestions:**

Typo: Figure 3, MultiBOS-AFL -> MBOS-AFL?

**Other Strengths And Weaknesses:**

S1. The paper is mostly clearly written.
S2. The paper offers analysis and hence justification for the proposed bidding method, which is technically sound.
S3. The empirical results are promising against other baseline bidding strategies, although existing methods are limited because the literature is scarce.

W1. More motivation of why multi-session auction scenario is realistic seems to be helpful.
W2. The paper would benefit from a clearer exposition of its assumptions regarding data owner  behaviors. While the authors describe the decision decision-making process of data consumers, it's unclear whether these are based on empirical observations, theoretical models, or simplifying assumptions. A more explicit statement of the behavioral assumptions underpinning the model would strengthen the paper's foundation and help readers better understand the scope and limitations of the proposed approach.
W3. It is suggested to discuss the technical challenges of designing proposed mechanisms.

**Questions For Authors:**

What bidding strategies did the other bidders use in the experiments? Were their approaches similar or distinct?

**Relation To Broader Scientific Literature:**

The work extends DC-oriented AFL by addressing multi-session budget pacing, a gap in prior single-session methods (e.g., Fed-Bidder, RLB). It connects hierarchical RL [Ref 1] to AFL, offering a novel integration.
[Ref 1]  Pateria, S., Subagdja, B., hwee Tan, A., and Quek, C. Hier archical reinforcement learning: A comprehensive survey. ACM Computing Surveys, 54(5):109:1–109:35, 2021

**Theoretical Claims:**

No issue found with regard to theoretical claims

---

> ### Author Rebuttal · Authors · 2025-04-01
>
> Thank you for your insightful and encouraging feedback. Below, we provide detailed, point-by-point responses to the key questions raised in the comments.
>
> >W1. More motivation of why multi-session auction scenario is realistic seems to be helpful.
>
> Practical federated learning environments typically require sequential collaboration phases where the collective model evolves gradually through repeated information consolidation and parameter adjustments. Consider healthcare applications: medical facilities (serving as information contributors) supply clinical data incrementally across various timeframes, leading to progressive enhancement of the shared analytical framework.
>
> The operational reality demonstrates that participating entities join the collaboration asynchronously, while available funding resources may vary throughout the project lifecycle. This dynamic creates an ideal application for multi-round bidding structures. Such an approach accurately reflects the continuous nature of knowledge development in these collaborative ecosystems, where both informational assets and processing capabilities are distributed through successive competitive allocation events.
>
> >W2. The paper would benefit from a clearer exposition of its assumptions regarding data owner  behaviors. While the authors describe the decision decision-making process of data consumers, it's unclear whether these are based on empirical observations, theoretical models, or simplifying assumptions. A more explicit statement of the behavioral assumptions underpinning the model would strengthen the paper's foundation and help readers better understand the scope and limitations of the proposed approach.
>
> The method we've developed operates without assumptions regarding the other types of stakeholders' behaviors. Rather, it features inherent adaptability to accommodate diverse stakeholder actions across multiple contexts. Our outlined decision protocols maintain flexibility and remain independent of underlying models, enabling effective functioning regardless of participant motivations or strategic choices. This versatility represents a fundamental advantage of our approach, as it facilitates widespread implementation across various aggregated federated learning environments without limitations imposed by strict behavioral assumptions or predefined conduct expectations.
>
> >W3. It is suggested to discuss the technical challenges of designing proposed mechanisms.
>
> The first technical challenge lies in optimizing the computational efficiency of utility estimation. Our methodology implements a framework combining Shapley-derived valuations with beta-based reputation tracking to quantify the utility a DC can gain from each DO. However, while Shapley-based methods are conceptually straightforward, they are computationally intensive. Additionally, another challenge is ensuring robust winning price modeling across diverse data owner distributions, as well as balancing the bidding strategy’s performance with budget constraints. Addressing these complexities is crucial for adapting our approach to practical AFL market environments, and we will add further details on these points in the revised paper.
>
> >Typo
>
> Thank you for pointing that out. We have corrected the typo in the revised manuscript.
>
> > What bidding strategies did the other bidders use in the experiments? Were their approaches similar or distinct?
>
> As detailed in the experiment section, we compared diverse advanced bidding strategies such as FBs, FBc, among others, by various bidders. This setting is designed to address concerns regarding the potential impact of competitors with advanced bidding strategies. However, if all bidders adopt the same bidding strategy, the differences in their individual historical data could lead to variations in their utility estimation and winning price models. This variance might result in differences in their bidding behaviors. Moreover, in practice, different bidders target diverse data owners due to their distinct training tasks. These factors influence their bids through specific parameters and model approaches in their bidding strategies. Nevertheless, it remains an intriguing area for future research to analyze the dynamics and potential equilibria that arise when all data consumers adopt the same strategy.

---

> > ### Comment · Reviewer_KgPU · 2025-04-07
> >
> > I have read the rebuttals from the authors, which have addressed most of my previous concerns.
> > Thus I decide to remain my score unchanged.

---

### Official Review · Reviewer_wy2c · 2025-03-11

**Overall Recommendation:** 3

**Summary:**

This paper introduces MBOS-AFL, a hierarchical reinforcement learning-based strategy for multi-session budget optimization in forward auction-based federated learning (AFL). The key idea is to enable data consumers (DCs) to dynamically allocate budgets across multiple FL training sessions (via an inter-session pacing agent) and optimize per-session bidding (via an intra-session agent). The method aims to maximize utility, defined as the aggregated reputation of recruited data owners (DOs). Experiments on six FL benchmarks (MNIST, CIFAR-10, etc.) demonstrate that MBOS-AFL outperforms seven baselines. The author addressed my concerns and I kept my score for technical contribution perspectives ## update after rebuttal

**Claims And Evidence:**

The claims are convincingly supported by experiments across diverse settings, including IID/Non-IID data and noisy conditions. The consistent outperformance of MBOS-AFL over RLB and Fed-Bidder variants highlights its novelty in multi-session pacing.

**Essential References Not Discussed:**

All relevant AFL and RL works appear to be cited.

**Experimental Designs Or Analyses:**

The experimental design is extensive with five budget settings and multiple experimental scenarios.

**Methods And Evaluation Criteria:**

The hierarchical RL design effectively decouples pacing and bidding decisions. Benchmark datasets and realistic noise/heterogeneity settings ensure relevance to real-world FL challenges.

**Other Comments Or Suggestions:**

1) In page 7 (Figure 3) "MultiBOS-AFL" seems shall be "MBOS-AFL"; the accuracy plots could benefit from clearer legends
2) In page 8 (Table 2), missing the percentage sign

**Other Strengths And Weaknesses:**

Strength:

1) The paper’s contents are well-organized.
2) The proposed method is technique soundness.
3) The paper provides insight analysis through its analysis and comparison experiments.
4) The paper proposes a new benchmark dataset processing for auction-based federated learning.

Weakness:

1) The system uses the SPSB auction mechanism, why is not bidding the value truthfully optimal? That is the main promise of this type of auction.
2）How to make sure the utility estimation is accurate?
3) The experiments could be enhanced by including some auction-based datasets or data from AFL market.
4) How practical is the proposed framework for real-world use?

**Questions For Authors:**

refer to the Weakness

**Relation To Broader Scientific Literature:**

This paper studies bidding strategy for data consumers in AFL, advancing bidding strategies beyond data privacy concerns.

**Theoretical Claims:**

No theoretical claims are made; the work is empirically driven.

---

> ### Author Rebuttal · Authors · 2025-04-01
>
> Thank you for your encouraging and insightful feedback. Below, we provide point-by-point explanations to key questions raised in the comments.
>
> >W1): The system uses the SPSB auction mechanism, why is not bidding the value truthfully optimal? That is the main promise of this type of auction.
>
> In the generalised second price auctions, auction theory [1] proved that bidding the true value of the data resources is optimal towards maximizing the accumulated utility. However, the authors in [2] demonstrated that truthful bidding might not be optimal under budget constraints and limited availability of data resources. In this sense, it is necessary to design bidding strategies for data consumers to automatically calculate bid prices for the available data resources in view of the budget limit, instead of bidding with the prices in the simplified linear form.
>
> >W2）How to make sure the utility estimation is accurate?
>
> Accurate utility estimation is crucial for the proposed. In practice, it is recommended to continuously monitor and update the utility estimation model as new bidding data becomes available, ensuring its adaptation to potential changes in the AFL environment.
>
> >W3) The experiments could be enhanced by including some auction-based datasets or data from AFL market.
>
> Thanks a lot for your suggestion. Since publicly available data from the AFL market is not accessible, and existing auction-based datasets, such as those from online advertising, are not directly applicable to a federated learning setting, we have followed established methods [3,4] to collect and generate the datasets used in our experiments. We will also explore additional auction-based datasets that could potentially be adapted for a federated learning setting.
>
> >W4) How practical is the proposed framework for real-world use?
>
> Indeed, the proposed method can be adapted for use in most FL settings that include an incentive mechanism with data consumers competing to bid for data owners [5,6,7], particularly in auction-based FL scenarios. A practical real-world application is in gas usage estimation within the power generation and delivery industry [8].
>
> >1) In page 7 (Figure 3) "MultiBOS-AFL" -> "MBOS-AFL"; the accuracy plots could benefit from clearer legends
> >2) In page 8 (Table 2), missing the percentage sign
>
> Thanks for your correction and suggestions. We have revised the typos in the manuscript and revised the legend of the accuracy plots to make it clear following your suggestions.
>
> -----
>
> [1] Vijay Krishna. Auction theory. Academic press, 2009.
>
> [2] Weinan Zhang et al. Optimal real-time bidding for display advertising. In KDD, 2014.
>
> [3] Yutao Jiao et al. Toward an automated auction framework for wireless federated learning services market. TMC, 2020.
>
> [4] Rongfei Zeng et al. Fmore: An incentive scheme of multi-dimensional auction for federated learning in MEC. In ICDCS, pages 278288, 2020.
>
> [5] Yutao Jiao et al. Toward an automated auction framework for wireless federated learning services market. TMC, 2020.
>
> [6] Rongfei Zeng et al. Fmore: An incentive scheme of multi-dimensional auction for federated learning in MEC. In ICDCS, pages 278288, 2020.
>
> [7] Palash Roy et al. Distributed task allocation in mobile device cloud exploiting federated learning and subjective logic. Journal of Systems Architecture, 113(2):doi:10.1016/j.sysarc.2020.101972, 2021
>
> [8] Hao Sun et al, "HiFi-Gas: Hierarchical Federated Learning Incentive Mechanism Enhanced Gas Usage Estimation," in IAAI-24, 2024.

---

### Official Review · Reviewer_1uYb · 2025-03-12

**Overall Recommendation:** 4

**Summary:**

The authors deal with the multi-session budget allocation problem for data consumers in auction-based federated learning and propose the MBOS-AFL by introducing a a hierarchical RL framework.

**Claims And Evidence:**

claims are well-supported.

**Essential References Not Discussed:**

No major omissions.

**Experimental Designs Or Analyses:**

The experimental setup is sound, with detailed descriptions of datasets, baselines, and training protocols.

**Methods And Evaluation Criteria:**

The method is novel and appropriate for multi-session AFL.

**Other Comments Or Suggestions:**

1. Several citations are missing publication years or have unclosed brackets, like "Tang and Yu ()"（Page 2, Section 3).

2. Table 1’s formatting could be improved for readability

**Other Strengths And Weaknesses:**

The research problem is important and the proposed approach is clear and technique soundness. However,

1. The selected baselines are outdated to some extent. It is suggested to incorporate more recent ones like [1, 2] with the same setting to help position the proposed method. In addition, Can you compare your bidding strategy in auctions with that of [3], though it is not designed for FL settings, it seems quite relevant. Also, can you comment on peer-based mechanisms (e.g., [4],[5],[6] or the references cited therein) vs auction-based mechanisms?


2. Limited exploration of communication costs in FL training rounds.

[1] Tang X, Yu H, Li Z, et al. A bias-free revenue-Maximizing bidding strategy for data consumers in auction-based federated learning[C]//Proc. IJCAI. 2024.

[2] Tang X, Yu H. Competitive-Cooperative Multi-Agent Reinforcement Learning for Auction-based Federated Learning[C]//IJCAI. 2023: 4262-4270.

[3] Chandlekar, Sanjay, et al. "Multi-unit Double Auctions: Equilibrium Analysis and Bidding Strategy using DDPG in Smart-grids." Proceedings of the 21st International Conference on Autonomous Agents and Multiagent Systems. 2022.

[4] Richardson, Adam, Aris Filos-Ratsikas, and Boi Faltings. "Budget-bounded incentives for federated learning." Federated Learning: Privacy and Incentive (2020): 176-188.

[5] Witkowski, Jens, and David C. Parkes. "A robust bayesian truth serum for small populations." Twenty-Sixth AAAI Conference on Artificial Intelligence. 2012.

[6] Radanovic, Goran, Boi Faltings, and Radu Jurca. "Incentives for effort in crowdsourcing using the peer truth serum." ACM Transactions on Intelligent Systems and Technology (TIST) 7.4 (2016): 1-28.】

**Questions For Authors:**

1: How does MBOS-AFL perform in scenarios with competing DCs using similar strategies?

2: Could the method be extended to handle non-static budget constraints (e.g., dynamically changing total budgets)?

  3: How does MBOS-AFL handle abrupt changes in DO availability across sessions?

**Relation To Broader Scientific Literature:**

This work advances AFL by addressing the multi-session budget pacing gap while integrating reputation systems and hierarchical RL.

**Theoretical Claims:**

This paper focuses on problem and algorithmic innovation, and empirical validation

---

> ### Author Rebuttal · Authors · 2025-04-01
>
> We sincerely appreciate your thoughtful suggestions and valuable feedback.
>
> >1. The selected baselines are outdated to some extent. It is suggested to incorporate more recent ones like [1, 2] with the same setting to help position the proposed method. In addition, Can you compare your bidding strategy in auctions with that of [3], though it is not designed for FL settings, it seems quite relevant. Also, can you comment on peer-based mechanisms (e.g., [4],[5],[6] or the references cited therein) vs auction-based mechanisms?
>
> Both [1] and [2] propose bidding strategies for DCs in the AFL marketplace, but with different objectives. While [1] focuses on revenue maximization, differing from the utility maximization in this paper, the formulations are distinct. [2], however, aims to maximize utility and maintain the AFL marketplace's health, using a reinforcement learning approach similar to the RLB baseline method in this paper.
>
> Both [3] and this paper focus on designing bidding strategies. However, different from [3] which aims to reach an equilibrium between buyers and sellers by designing bidding strategies for both buyers and sellers, we focus on guiding DCs (buyers) to bid for DOs (sellers) by budget management strategies to maximize their KPIs within a given budget. This difference in overall design goals makes it unsuitable to compare [3] (similarly [4][5][6]) with the proposed approach.
>
> >2. Limited exploration of communication costs in FL training rounds.
>
> The proposed MSBO-FL framework consists of two main parts: the multi-session budget optimization mechanism (i.e., DO recruitment process) and the FL model training (described in “Preliminaries”).
>
> In the DO recruitment part, computational complexity primarily arises from the utility estimation function and hierachical reinforcement learning (HRL)-based budget allocation phase. The HRL-based budget allocation is with a time complexity of $O(|S|·d + |C_s|·d + m)$, where $|S|$ represents the state space size, $|C_s|$ denotes the number of available qualified DOs in session $s$, $d$ is the neural network dimension, and $m$ is the minibatch size used in the training procedure.
>
> For the utility estimation, the computational demand largely stems from the Shapley value-based contribution evaluation. We employ the Beta Reputation System combined with Shapley Value technique, which has a computational complexity of $O(T·N·log N)$, where $T$ and $N$ are the number of training rounds and data owners, respectively. Efficient contribution evaluation remains a challenging issue in federated learning, where high time complexity is common in related problems.
>
> In the FL model training part, we adopt the FedAvg method, which is widely used in FL research. To compare the computational complexity of different bidding methods for data consumers in AFL, we can focus on the DO recruitment part, as the FL model training generally relies on FedAvg across methods. This recruitment part typically involves utility estimation and other functions, with the distinguishing feature of our proposed MBOS-AFL method being the two-level HRL-based budget allocation that optimizes both inter-session budget pacing and intra-session bidding strategies.
>
> >Other Comments Or Suggestions
>
> Thanks a lot for your correction. We have revised the manuscript following your suggestions.
>
> >Q1: How does MBOS-AFL perform in scenarios with competing DCs using similar strategies?
>
> If all DCs adopt the same bidding strategy, the differences in their individual historical data could lead to variations in their utility estimation and their states. This variance might result in differences in their bidding behaviors (i.e., actions). Moreover, in practice, different DCs target diverse data owners due to their distinct training tasks. These factors influence their bids through specific parameters and model approaches in their bidding strategies. Nevertheless, it remains an intriguing area for future research to analyze the dynamics and potential equilibria that arise when all data consumers adopt the same strategy.
>
> >Q2: Could the method be extended to handle non-static budget constraints (e.g., dynamically changing total budgets)?
>
> MBOS-AFL framework be naturally extended to accommodate dynamically changing total budgets with minimal modifications. The current design already incorporates state representations that track remaining budgets and historical allocations, making it well-positioned to handle budget fluctuations.
>
> >Q3: How does MBOS-AFL handle abrupt changes in DO availability across sessions?
>
> The changes in DO availability have limited impact on the operation of MBOS-AFL. This is because such changes directly influence the state observed by the bidding agents, which then adjust their actions based on the learned policy. As a result, MBOS-AFL can dynamically generate appropriate bids in response to variations in DO availability, maintaining robust and adaptive performance under changing conditions.

---

> > ### Comment · Reviewer_1uYb · 2025-04-06
> >
> > Thank you for the author's response. After reading the author's rebuttal, the main concerns I had have been addressed, and I will maintain my score.

---

### Decision · Program_Chairs · 2025-05-01

**Decision:**

Accept (poster)

**Comment:**

The paper is well-organized with clear motivation and method. The proposed method is novel. Its claims are well supported by theoretical and experimental analysis. The suggested refinement would be justify how the proposed scenario is aligned to real applications, for example, using dataset from real applications, and provide a better justification.